# Methane and Dissolved Organic Matter in the Ground Ice Samples from Central Yamal: Implications to Biogeochemical Cycling and Greenhouse Gas Emission

**Petr B. Semenov [1,*], Anfisa A. Pismeniuk [1,2,*], Sergei A. Malyshev [1], Marina O. Leibman [3,4], Irina D. Streletskaya [2], Elizaveta V. Shatrova [1], Alexander I. Kizyakov [2] and Boris G. Vanshtein [1]**

[1] All-Russia Institute for Geology and Mineral Resources of the World Ocean (VNIIOkeangeologia), 190121 Saint-Petersburg, Russia; serg.i-karamba@yandex.ru (S.A.M.); shatliza@yandex.ru (E.V.S.); vanshbor@mail.ru (B.G.V.)

[2] Department of Cryolitology and Glaciology, Faculty of Geography, Lomonosov Moscow State University, 119991 Moscow, Russia; irinastrelets@gmail.com (I.D.S.); akizyakov@mail.ru (A.I.K.)

[3] Tyumen Scientific Centre SB RAS, Earth Cryosphere Institute, ulitsa Malygina, 86, P.O. Box 1230, 625000 Tyumen, Russia; moleibman@gmail.com

[4] Laboratory of Polar and Sub-Polar Geosystems, Institute X-BIO, University of Tyumen, ulitsa Volodarskogo, 6, 625003 Tyumen, Russia

* Correspondence: petborsem@gmail.com (P.B.S.); apismeniuk@gmail.com (A.A.P.)

**Abstract:** Permafrost thawing leads to mobilization of the vast carbon pool into modern biogeochemical cycling through the enhanced release of dissolved organic matter (DOM) and production of greenhouse gases ($CO_2$ and $CH_4$). In this work, we focus on the study of methane and DOM distribution and genesis in the ground ice samples of thermodenudational exposure in the Central Yamal (Russian Arctic). We propose that the liberation of the ice-trapped $CH_4$ and generation of $CO_2$ by DOM mineralization are the earliest factors of atmospheric greenhouse gases emission as a result of permafrost thawing. The observed enormously "light" isotope signatures of methane ($\delta^{13}C$ < −80‰, $\delta D$ < −390‰) found in the tabular ground ice units significantly divergent in morphology and localization within the exposuremay be related to subzero (cryogenic) carbonate reduction a as significant factor of the local methane enrichment. DOM is mainly formed (>88%) by biochemically refractory humic acids. Distribution of the labile protein-like DOM reflects the specific features of carbon and nitrogen cycles in the tabular ground ice and ice wedge samples. Tabular ground ice units are shown to be a significant source of methane and high quality organic matter as well as dissolved inorganic nitrogen (DIN). Ice wedges express a high variation in DOM composition and lability.

**Keywords:** permafrost; biogeochemical cycling; climate change; greenhouse gas emission; dissolved organic matter; ground ice

## 1. Introduction

The ongoing climate change promotes the processes of cryogenicnature in the permafrost-affected landscapes [1]. The most important issue of the permafrost thawing is the mobilization of the vast carbon pool into modern biogeochemical cycling. Permafrost carbon reservoir in the northern hemisphere is evaluated as 1832 PgC (1 Pg = 1015 g) [2] which is about 2.5 times as much as that in the current atmosphere [3]. Dissolved organic carbon (DOC) is the most mobile fraction of C easily transported by diffusion or advective fluxes of dissolved organic matter (DOM) species [4,5]. Permafrost-derived DOM contains low molecular weight substrates that have been enzymatically

processed for biological consummation within the soil paleoenvironment before being physically locked by freezing. The accelerated turnover of the DOM-linked carbon pool is proposed to be responsible for a positive feedback loop of climate warming due to amplified greenhouse gas (GHG) generation. The most important GHG related to the microbially mediated carbon cycle is carbon dioxide ($CO_2$) produced by both aerobic respiration and anaerobic fermentation processes and methane ($CH_4$) formed in strictly anaerobic conditions [6].

Labile organic matter (OM) starts to degrade immediately after frozen deposits thaw and the corresponding biodegradable DOC (BDOC) is rapidly converted to $CO_2$ within the hydrological network (if thaw water is exported as water streams) or within the thawed soil (if thaw water is retained) [7]. Short time incubation tests demonstrated that ca 50–53% of > 20,000-year-old BDOC stored in permafrost soil was utilized by heterotrophic bacteria for aerobic respiration within a week of incubation at 20 °C [7–9]. The 7 year incubation has shown that stable production of $CH_4$ by archaeal community started after a pronounced lag phase of a 4 year duration. Therefore, the rate of methanogenesis was limited by slow multistep processing of biochemically refractory high-molecular-weight DOM yielding the appropriate substrates for carbon reduction. In terms of GHG potential (expressed in $CO_2$ equivalents, $CO_2$-Ce) the contributions from the experimentally produced $CO_2$ and $CH_4$ were similar, although a much larger carbon pool was mineralized to $CO_2$ during the incubation [10]. The numerical model calibrated against the experimental data predicts a higher loss of permafrost carbon under oxic conditions ($113 \pm 58$ g $CO_2$–C kgC$^{-1}$ (kgC, kilograms of carbon) by 2100 (90 years), but the resulting $CO_2$–Ce production is going to be twice as high ($241 \pm 138$ g $CO_2$–Ce kgC$^{-1}$) under anoxic conditions [10].

Methane trapped in frozen sediments was estimated to be minor in overall contribution compared to microbial methane generation within the permafrost zone [11]. However, immediate volatilization of the physically immobilized methane upon permafrost thawing is likely a factor of the fast impact of permafrost thawing on atmospheric greenhouse gas (GHG) composition. Ground ice of the Yamal Peninsula has been reported to express drastic variations in $CH_4$ content with mean values around 500 ppmV up to a maximum of 9182 ppmV recorded in a tabular ground ice (TGI) sample [11]. Stable isotope composition of TGI–derived $CH_4$ was characterized by $\delta^{13}C$ values ranging from −62 to −74‰ and $\delta D$ values ranging from −259 to −330‰ [12]. Given the lack of diffusion and advection, the specific transport mechanisms such as cryogenic displacement have been shown to move methane-enriched zones a distance of several meters [13]. Another explanation of uneven methane distribution in permafrost units is in-situ methanogenesis which has been proved to operate in subzero conditions, but it is not well understood in terms of overall significance for permafrost carbon cycling [13,14]. The enhanced isotope fractionation leading to the appearance of extremely low values of $\delta^{13}C$ (<−80‰, PDB) may point at the contribution from subzero methane formation, while the increased values of $\delta^{13}C$ coupled to an elevated amount of $C_{2+}$ hydrocarbon gases are indicative of post-production isotope and molecular fractionation processes in methane cycle [14,15]. Based on the published data, we propose that the liberation of the ground ice-trapped $CH_4$ and generation of $CO_2$ by aerobic respiration of the labile DOM are the processes providing the immediate emission of GHG as a result of permafrost thawing. While the stable CH4 production from organic matter delivered by the denudational mudflows into the adjacent anaerobic waterlogs will start much later.

Thermodenudation, developed within a coastal zone of tundra lakes, provides perfect conditions for monitoring the ground ice-derived organic matter biogeochemical transformation. It starts from a thaw-induced detachment of source material and continues throughout downslope transportation and discharge into the adjacent lake and mixing with lake water. The cryogenic processes linked to thermocirques formation and dynamics have been thoroughly described in the study area (Vaskiny Dachi Research Station, Central Yamal) [16,17]. An export of colored dissolved organic matter (CDOM) into adjacent lakes as a result of thermodenudation has been recently shown as the 5–7-fold lake water dissolved carbon enrichment relative to the background. CDOM is represented mainly by the biochemically refractory allochthonous DOM: humic and fulvic acids [18]. Data on

labile autochthonous DOM composition and biochemical transformation is strongly required for understanding the biogeochemical cycling linked to the ground ice formation and evolution [19].

Based on the above review, in this work, we focus on studying the distribution, composition, and genesis of CH$_4$ and biochemically available DOM (labile) fraction in the ground ice deposits sampled from the exposed frozen wall of thermocirque located in the center of Yamal Peninsula. The assessment of heterogeneity of biogeochemical environments within the sampled exposure, primarily governed by the ground ice origin (tabular ground ice and ice wedges), is one of the major issues of our work. Using the protein-like DOM fluorescent intensity as a biomarker of labile DOM we may observe the characteristic signatures of both carbon and nitrogene cycles [19].

## 2. Study Area

The key site (Vaskiny Dachi Research station, 70°17′ N, 68°54′ E) is located at the watershed of Se-Yakha and Mordy-Yakha rivers, in the central part of the Yamal Peninsula (Figure 1). The Bovanenkovo oil and gas condensate field, located 30 km to the north, was commissioned in 2012. The anthropogenic impact provokes an additional activation of destructive cryogenic processes, which requires geochemical studies of permafrost. Particular interest in hazardous cryogenic phenomena in the study area raised after the formation of the gas emission crater in 2013, approximately 40 km to the south of the Bovanenkovo oil and gas condensate field [20,21].

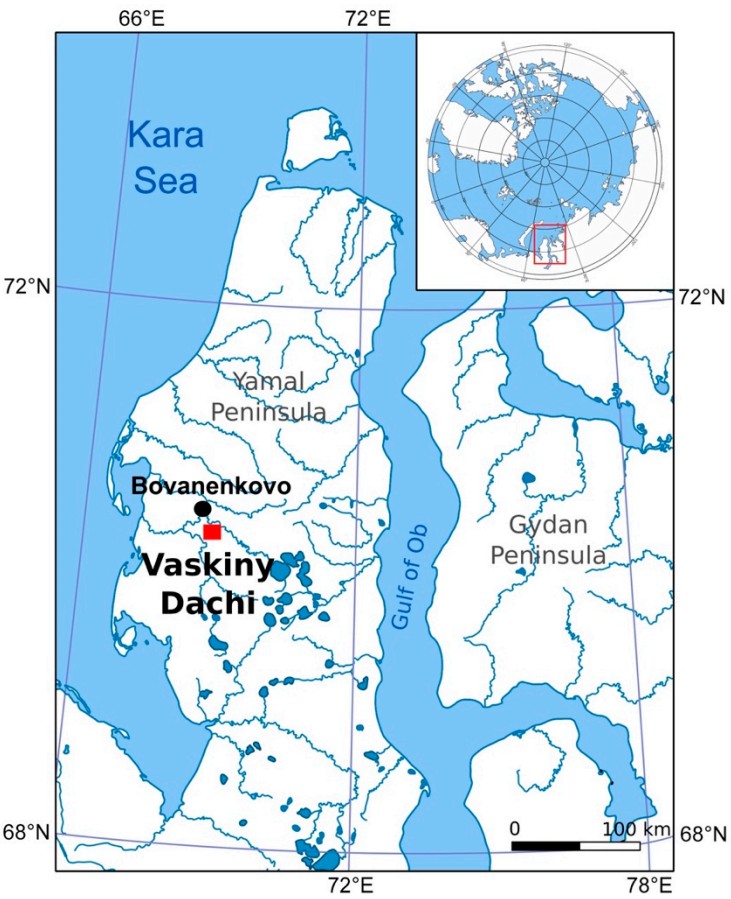

**Figure 1.** Study area map showing the location of the study key site.

The central part of the Yamal Peninsula is located in the zone of continuous permafrost. Permafrost depth (0 °C isotherm) in the study area reaches 300 m, the maximum depth of ice-bonded permafrost is 130 m [22]. Active layer depth (ALD) varies between 0.40 and 1.20 m [16]. In the study area, massive ground ice (TGI and ice wedges) is widespread, occurring close to the surface. Cryogenic slope

processes are especially active on the territory [23]. Climate change and active anthropogenic impact of the region influenced the activation of cryogenic processes [16]. The abnormally warm summer of 2012 provoked the thawing of the upper part of the ice-rich permafrost, including TGI and ice wedges. Massive descents of retrogressive thaw slumps lead to the formation of thermocirques [17]. The study of Quaternary deposits and ground ice from the wall (Figure 2) of the newly formed thermocirque served as the basis for this article.

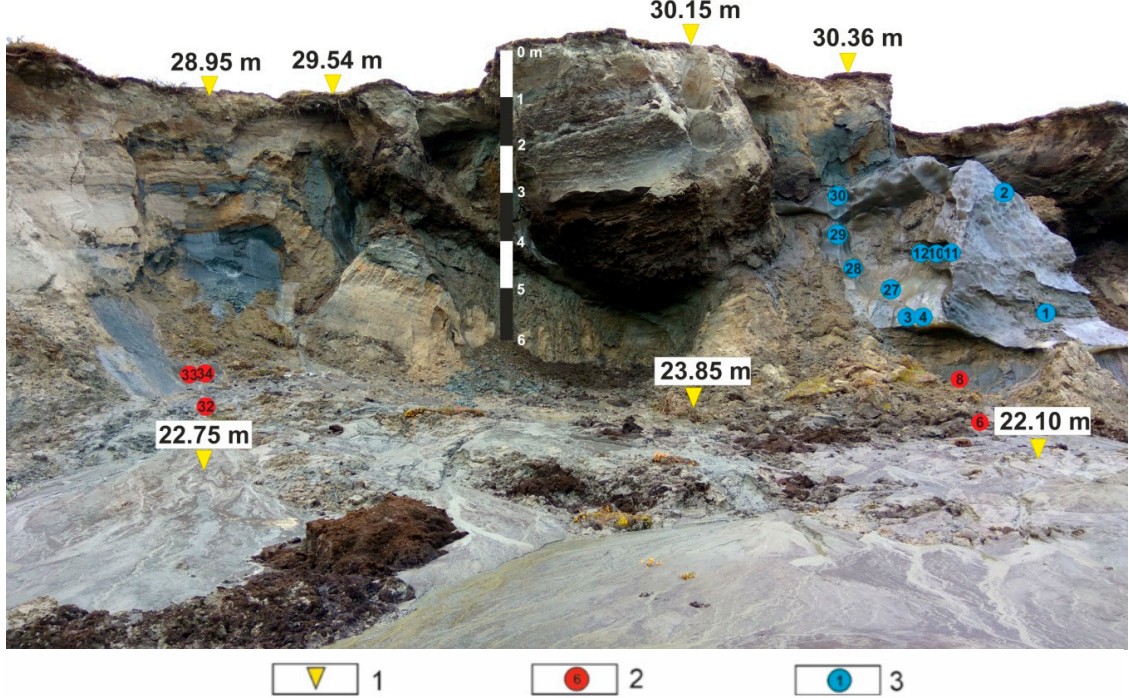

**Figure 2.** Location of sampling points in thermocirque at Vaskiny Dachi (photo by A.I. Kizyakov [24]). Legend: **1**—elevation (meters above sea level); **2**—location and number of tabular ground ice (TGI) sample; **3**—location and number of ice-wedge sample.

I.D. Streletskaya and M.O. Leibman published a geological section with cryopeg lenses and TGI of the Central Yamal [25]. Sand and clay deposits (MIS 5–MIS 3), often containing thick (up to 40 m) TGI, represent the unit of the Quaternary sediments. TGI overlies the sequence of sand and silty sand with a thickness of 30–40 m, underlain by clay deposits at the base of the section (MIS 6). The lenses of cryopegs are discovered at different depths in the sands. Holocene (MIS 1) slope deposits (sIV) represent the upper part of the studied section (Figure 3). Below lies a sequence of continental sediments (aIII$^{2-4}$) with Late Pleistocene ice wedges (MIS 3–MIS 2), characterized by an isotopic composition of ice and deuterium excess typical for ice wedges (average values: δ$^{18}$O = −22‰, δD = −165‰, d$_{excess}$ = 11‰ [24]). The host sediments are sandy loam with peat lenses (bIII). Published [24] radiocarbon dates of peat confirm that it was formed during MIS 3 (30,900 ± 1300 cal yr BP, 32,200 ± 1300 cal yr BP; 37,650 ± 1950 cal yr BP). Ice wedges penetrate the underlying TGI, which is often found in permafrost sections of Quaternary deposits in the north of West Siberia. The lower parts of the ice wedges, penetrating TGI, were formed epigenetically, and the upper ones-syngenetically. The origin of TGI is disputed [25–27]. I.D. Streletskaya and M.O. Leibman [25] approve a cryogeochemical model of epigenetic freezing of marine sandy-clay sediments, accompanied by the TGI formation at the sand-clay interface and occurrence of cryopegs. In their opinion, the TGI formed during the freezing of water-saturated sediments immediately after marine regression or in shoal conditions not later than MIS 3.

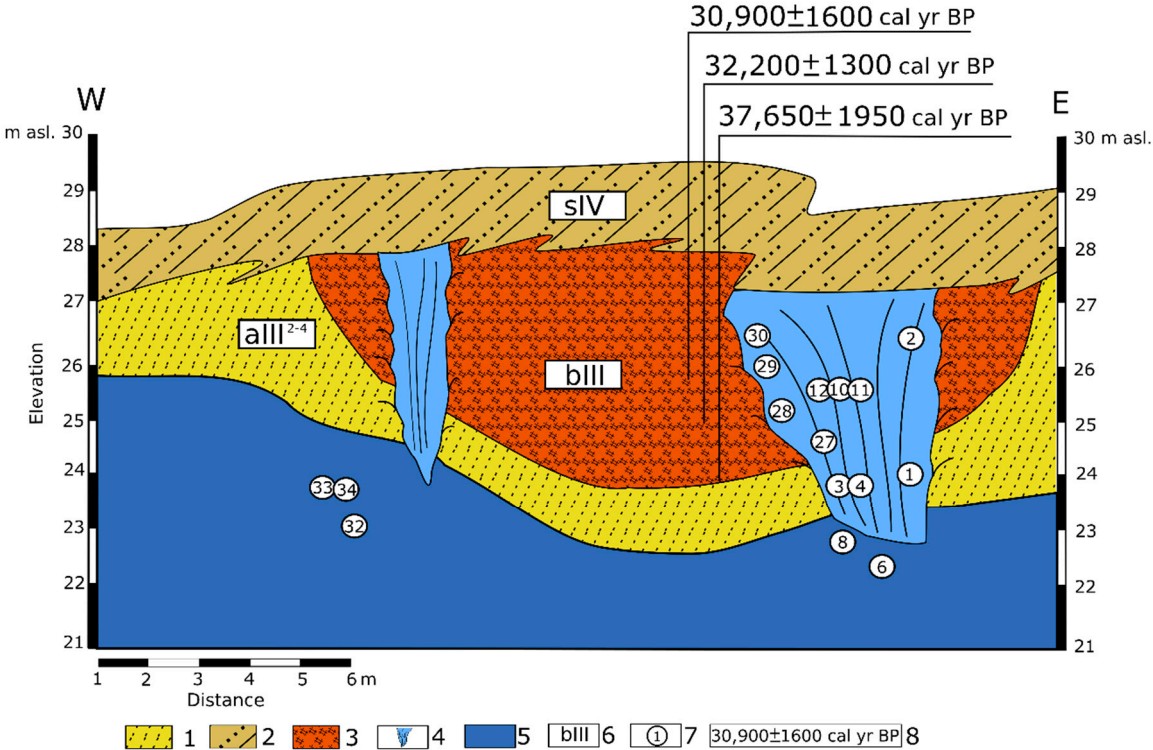

**Figure 3.** Schematic thermocirque section in the Vaskiny Dachi area. Legend: **1**—sandy loam; **2**—sand and sandy loam; **3**—peat; **4**—ice wedge; **5**—tabular ground ice; **6**—geological and genetic index of deposits; **7**—location and numbers of samples; **8**—calibrated $^{14}$C dates of deposits [24].

## 3. Materials and Methods

The ground ice samples were stored frozen (at −20 °C) until laboratory analysis in the analytical center of Federal State Institution "VNIIOkeangeologia" (Saint-Petersburg, Russia).

### 3.1. Ground Ice Physical Properties and Dissolved Gas Analysis

The ground ice samples of about 1 kg were sawn lengthwise. The resulting rectangular fragments were immediately weighted in a cold room. The weight of the fragments ranged within 45–50 g. One set of the fragments was used for the analysis of bulk free gas content. For this purpose, the ground ice fragments were placed into the specialized plastic bags, which were then immediately evacuated with the pump unit, so that no air space was left between the sample and the plastic bag. The sealed samples were left to thaw at +4 °C. After the complete thawing, the free gas liberated from the inclusions was collected in the graduated cylinder by piercing the plastic bag submerged in the vessel of saturated NaCl solution and capturing the released bubbles with an upside-down funnel. The volume of the captured gas was recorded at atmospheric pressure and the bulk free gas content was calculated for the weight of the ground ice sample (bulk free gas, cm$^3$/kg). Another rectangular fragment of the ground ice sample was placed in the 250 mL argon-flushed crimped vial for the static headspace analysis [28] of C$_1$–C$_5$ hydrocarbon gases (HG). The sealed headspace vials with ground ice samples were agitated with orbital shaker Heidolph Unimax 2010 for 2 h at a speed of 400 rpm. The equilibrium gas sample was introduced into the injector of Shimadzu 2014 gas chromatograph (GC) using the 10-port gas valve (Valco) with a 1 cm$^3$ evacuated gas loop. Restek Rt-Aluminia BOND/Na$_2$SO$_4$ wide-bore capillary column (i.d. 0.53 mm, length 50 m, film thickness 10 μm) attached to the packed injector and flame ionization detector (FID) was used for analysis. Helium was used as a carrier gas at a flow rate of 25 mL/min. Certified gas mixtures were used as external standards. The detection limit of the analysis was 50 ppb, for CH$_4$. The uncertainty of the instrumental measurements was ± 2.5%. HG concentrations (ng/g) were calculated using the partial pressure and Bunsen solubility coefficients [29]. Total headspace

gas was calculated as the sum of the HG $C_1$-$C_5$ mixing ratio (ppm) of the gas wetness (gas wet percent, wetness coefficient, kW) was determined according to the formula $\Sigma C_2$–$C_5/\Sigma C_1$–$C_5 \times 100$ [30].

The suspensions left from the analysis of free bulk gas and headspace analysis were combined for the soil fraction measurements. The combined aliquots were centrifuged using Haraeus Megafuge 10 at 4500 rpm (60 min), the remaining precipitates were freeze-dried using Scanvac Coolsafe 1100 and weighted. The resulting soil content was calculated as g/kg.

### 3.2. Stable Isotopic Composition of Carbon and Hydrogen in Methane

The isotope measurements were performed by isotopic-ratio gas chromatography/mass spectrometry (GC/IRMS). The minimum methane concentrations required for measurements were 20 and 200 ppmV for determination of $^{13}C/^{12}C$ and $^2H/^1H$, respectively. Carbon isotope composition ($^{13}C/^{12}C$) in $CH_4$ was measured with an Agilent 6890N GC (Agilent Technologies, Santa Clara, CA, USA) interfaced with a Finnigan Delta S IRMS (Bremen, Germany) using a Finnigan GC-C II interface. The GC was equipped with Molsieve column (12 m, 0.32 mm i.d.) and an injection valve. Samples were calibrated against a certified standard; the resulting isotopic signatures were reported in $\delta$ notation and per mil (‰) values vs. PDB ($\delta^{13}C$ PDB). Analyses were triplicated. The uncertainty for the measurements was greater than $\pm 0.2$‰.

The hydrogen isotope measurements ($^2H/^1H$) were performed on an Agilent 7890A GC (Agilent Technologies, Santa Clara, CA, US) was interfaced to a MAT 253 IRMS (Thermo Scientific, Bremen, Germany) using a GC-Isolink interface by Thermo. The GC was equipped with Molsieve column (12 m, 0.32 mm i.d.) and an injection valve. Samples were calibrated against a certified standard; the resulting isotopic signatures were reported in $\delta$ notation and per mil (‰) values vs. SMOW as ($\delta D$ SMOW). Analyses were triplicated. The uncertainty of the isotope measurements was $\pm 0.5$ ‰.

The liner regression equation based on both laboratory and field data from natural sulfate-depleted freshwater environments was used to evaluate the fractionation between $\delta D$-$CH_4$ on $\delta D$-$H_2O$ in the ground ice samples [31]:

$$\delta D\text{-}CH_4 = 0.675 \times (\delta D\text{-}H_2O) - 284‰ \ (p < 0.0005) \tag{1}$$

The hydrogen isotope fractionation factor between methane and methane formation water was calculated as [15]:

$$\alpha D = (\delta D H_2O + 10^3)/(\delta CH_4 + 10)^3 \tag{2}$$

### 3.3. Dissolved Organic Carbon (DOC) and Dissolved Inorganic Carbon (DIC) Measurements

Dissolved carbon species were measured using the Shimadzu TOC-V CSN element analyzer. Before analysis thaw water samples were transferred through a 0.45 μm glass microfiber syringe filter (GF/F) with a syringe. The uncertainty of the analytical measurements was ± 3%.

### 3.4. Thaw Water Solutes

The ion composition of thaw water samples was analyzed using Metrohm 940 Professional IC Vario ion chromatograph (IC) with a conductometry detector. For the separation of the anions ($NO^2$, $PO_4^{3-}$ $NO^{3-}$, $SO_4^{2-}$) we used Metrosepp A Supp 5–250/4.0 column and 5 mmoL $Na_2CO_3/NaHCO_3$ solution as an eluent at flow rate 1 mL/min. A chemical suppressor unit (MSM-A) was used for the reduction in background conductivity. For non-suppression IC ($NH_4^+$) we used column Metrosepp C6–250/4.0 and a mixture of 1.7 mM nitric and 1.7 mM dipicolinic acids solution as an eluent at a flow rate of 0.9 mL/min. The certified standard mixtures of ion composition (Fluka) were used for the method calibration and calculation of the resulting concentration values (mg/L). The uncertainty of the analytical measurements was $\pm 1.5$%. Dissolved inorganic nitrogen (DIN) was determined as a sum of $NO^{2-}$, $NO^{3-}$ and $NH_4^+$.

### 3.5. Fluorescence Measurements of Dissolved Organic Matter Molecular Composition

A Shimadzu RF5301 Fluoremeter was used for the measurement of the excitation-emission (EEM) fluorescent matrices within a range of 250–450 nm. The excitation and emission wavelengths steps were 2 and 1 nm, respectively. The inner filter effect was controlled by the sample dilution [32]. Spectral intensities were recorded by manual peak picking method in the 2D spectral peak extraction mode of the Shimadzu Panorama Fluorescence 3D 3.0 software. The identified peak maximums are indicated in Table 1. The intensity values were normalized to the Raman scattering peak area of the deionized water and expressed in Raman units (RU) [33]. The diagnostic spectral indices, fluorescent index (FI), and biological index (BIX) have been calculated [34].

**Table 1.** Summary and naming of the fluorescence peaks used for dissolved organic matter (DOM) molecular characterization [35,36].

| Ex300-370 Em400-500 | Ex237-270 Em400-500 | Ex270-280 Em300-320 | Ex270-280 Em320-350 |
|---|---|---|---|
| Humic-like, hDOM-1 | Humic-like, hDOM-1 | Protein-like (Tyrosine-like), TyrDOM | Protein-like (Tryptophan-like), TrpDOM |

### 3.6. Statistical Analyses

Statistical data processing was conducted using Statsoft Statistic 12 software. DOC and fDOM values were transformed by Box-Cox method before linear regression.

## 4. Results

### 4.1. Free Gas, Soil Content, and pH

Free gas content varied from 5.4 cm$^3$/kg (TGI sample #6) to 55.4 cm$^3$/kg (ice-wedge sample #2) at a mean value of 28.33 cm$^3$/kg (Table 2), thus indicating one order variation. The larger variation was encountered in the soil content, where the maximum value measured in the TGI sample #8 (844.6 g/kg), was four orders of magnitude higher than the minimum (0.01 g/kg) detected in the ice-wedge sample #2. The median amount of soil content was 2.92 g/kg. pH values varied from 8.36 (ice wedge #4) to 6.45 (TGI #12).

**Table 2.** Basic characteristics of the ground ice samples.

| Sample # | Sampling Depth, m | Bulk Free Gas, cm$^3$/kg | Soil, g/kg | pH |
|---|---|---|---|---|
| 1 | 4.26 | 27.1 | 1.40 | 7.1 |
| 2 | 4.26 | 55.4 | 3.80 | 7.1 |
| 3 | 6.01 | 42.6 | 36.80 | 7.32 |
| 4 | 6.01 | 34.2 | 0.04 | 8.36 |
| 10 | 5.06 | 40.2 | 57.64 | 6.96 |
| 11 | 5.06 | 37.5 | 1.22 | 6.76 |
| 12 | 5.06 | 27.4 | 0.01 | 6.45 |
| 27 | 5.46 | 19.6 | 0.02 | 6.9 |
| 28 | 5.01 | 29.3 | 4.62 | 6.41 |
| 29 | 3.91 | 24.0 | 12.44 | 7.42 |
| 30 | 3.46 | 47.7 | 0.33 | 7.55 |
| 6 * | 7.96 | 5.4 | 15.90 | 7.4 |
| 8 * | 7.96 | 8.8 | 844.65 | 7.5 |
| 32 * | 6.2 | 35.2 | 2.13 | 7.16 |
| 33 * | 4.9 | 14.5 | 0.43 | 7.08 |
| 34 * | 4.9 | 18.1 | 50.70 | 7.06 |
| Median value | | 28.3 | 2.92 | 7.1 |
| Std.dev. | | 13.9 | 208.94 | 0.5 |
| Coff.Var. | | 47.7 | 323.91 | 6.5 |

* tabular ground ice samples are marked with an asterisk, nd- not determined.

### 4.2. CH$_4$ Content, C$_2$–C$_5$ Gases

Methane concentrations ranged from 3.60 ng/g (ice-wedge sample #11) to 11,278 ng/g (TGI sample #33) with median value 55.9 ng/g (Table 3). Relatively high levels of dissolved CH$_4$ (above 1000 ng/g) appeared in three samples of TGI (#6, #8, #33) and one sample of ice-wedge ice (#3). Two TGI samples #33 and #8 may be considered as anomalous relative to the analyzed sample collection. The highest values of methane content in our samples were associated with lowered values of the free gas amount in TGI. Herewith, the only ice-wedge sample (#30) with CH$_4$ concentration <1000 ng/g, showed the free gas quantity (47.7 cm$^3$/kg) above the mean value (28.8 cm$^3$/kg). Stable isotope signatures of CH$_4$ ($\delta^{13}$C and $\delta$D) were measured in the samples of elevated gas content (Table 3).

**Table 3.** CH$_4$ concentrations and isotope composition, parameters of C$_1$–C$_5$ HG molecular composition.

| Sample # | CH$_4$, ng/g | $\delta^{13}$C$_1$, ‰ | $\delta$D$_1$, ‰ | Total HCG in HP, ppm | kW, % |
|---|---|---|---|---|---|
| 1 | 81.41 | - | - | 114 | 0.64 |
| 2 | 45.23 | - | - | 9 | 12.68 |
| 3 | 4.08 | - | - | 6 | 14.18 |
| 4 | 7.90 | - | - | 13 | 7.58 |
| 10 | 141.45 | - | - | 176 | 0.85 |
| 11 | 3.60 | - | - | 7 | 14.04 |
| 12 | 51.34 | - | - | 86 | 0.71 |
| 27 | 5.77 | - | - | 8 | 0.78 |
| 28 | 7.82 | - | - | 12 | 4.85 |
| 29 | 9.53 | - | - | 15 | 5.06 |
| 30 | 1282.30 | −68 | −378 | 1968 | 0.03 |
| 6 * | 1875.88 | −72 | −369 | 2880 | 0.04 |
| 8 * | 7890.47 | −84 | −394 | 9558 | 0.15 |
| 32 * | 78.72 | - | - | 20 | 4.21 |
| 33 * | 11,278.03 | −82 | −397 | 16,016 | 0.01 |
| 34 * | 60.59 | - | - | 90 | 1.55 |
| Median value | 55.96 | | | 53 | 1.19 |
| Std.dev. | 3287.34 | | | 4473 | 5.19 |
| Coff.Var. | 230.44 | | | 231 | 123.30 |

"-" not detrmined, * tabular ground ice samples are marked with an asterisk.

### 4.3. Bulk Geochemical Parameters (DOC, DIC, DIN), Dissolved Nutrients, and Electron Acceptors (NH$_4^+$, NO$_3^-$, SO$_4^{2-}$, PO$_4^{3-}$)

DOC concentrations varied from 4.89 mg/L in the gas-rich bubbly ice-wedge sample #27 to 21.42 mg/mL in the TGI sample #8 with the lowest free-gas gas level (8.8 cm$^3$/kg) at the highest soil content (844.6 g/kg). The median DOC value was 8.6 mg/L (Table 4). The observed DOC maximum reported for the Late Pleistocene ice wedge from the Siberian Arctic was 28.6 mg/L [37]. The maximum value of DOC in the thermocirque peat lens of the Central Yamal was as high as 243 mg/L [18].

DIC concentrations expressed more variability with a minimum of 0.69 mg/L in the ice-wedge sample #12 and a maximum of 16.37 mg/L in the ice-wedge sample #3. Median DIC concentration equaled 7.18 mg/L. The minimum DIN amount (0.3 mg/L) was encountered in the bubbly ice-wedge sample #12 (0.3 mg/L) which also showed the minimum DIC concentration. The maximum DIN was observed in the TGI sample #8 demonstrating the highest level of DOC. This sample represents the biogeochemical environment enriched in labile carbon pool, dissolved nutrients, and electron acceptors. Thaw water from TGI #8 shows the highest concentration of the nitrogen cycle species: NH$_4^+$ (7.08 mg/L), NO$_3^-$ (1.65 mg/L), and NO$_2^-$ (78 mkg/L) as well as PO$_4^{3-}$ (30.0 mkg/L) and SO$_4^{2-}$ (22.90 mg/L) [38]. The major contributor of DIN in both ice wedge and TGI samples was NH$_4^+$. The median SO$_4^{2-}$ concentration was 1.81 mg/L.

**Table 4.** Ground ice thaw water solutes concentration and bulk biogeochemical parameters.

| Sample # | $PO_4^{3-}$, mkg/L | $NO_3^-$, mg/L | $NH_4^+$, mg/L | $SO_4^{2-}$, mg/L | DIN, mg/L | DOC, mg/L | DIC, mg/L |
|---|---|---|---|---|---|---|---|
| 1 | 13.0 | 0.04 | 1.43 | 3.80 | 1.1 | 10.55 | 7.03 |
| 2 | 9.0 | 0.02 | 3.05 | 0.89 | 2.4 | 10.52 | 5.96 |
| 3 | 0.0 | 0.01 | 3.40 | 2.89 | 2.7 | 13.46 | 16.37 |
| 4 | 4.0 | 0.19 | 0.37 | 1.43 | 0.3 | 6.59 | 11.99 |
| 10 | 3.6 | 0.17 | 2.76 | 1.27 | 2.2 | 6.22 | 6.03 |
| 11 | 6.0 | 0.00 | 1.61 | 1.49 | 1.3 | 7.34 | 4.71 |
| 12 | 0.0 | 0.14 | 0.72 | 0.19 | 0.6 | 5.49 | 0.91 |
| 27 | 4.0 | 0.03 | 0.90 | 1.76 | 0.7 | 4.89 | 2.48 |
| 28 | 8.0 | 0.03 | 3.14 | 2.46 | 2.5 | 10.98 | 8.44 |
| 29 | 6.0 | 0.00 | 2.54 | 1.87 | 2.0 | 8.2 | 10.05 |
| 30 | 5.0 | 0.00 | 2.55 | 0.26 | 2.0 | 8.06 | 7.66 |
| 6 * | 8.0 | 0.32 | 1.61 | 8.35 | 3.2 | 13.34 | 9.9 |
| 8 * | 30.0 | 1.65 | 7.08 | 22.90 | 5.9 | 21.42 | 12.78 |
| 32 * | 6.0 | 0.01 | 1.98 | 5.93 | 1.5 | 6.58 | 3.08 |
| 33 * | 9.0 | 0.00 | 1.35 | 1.02 | 1.1 | 4.51 | 2.11 |
| 34 * | 8.0 | 0.04 | 3.00 | 2.06 | 2.3 | 9.65 | 7.33 |
| Median | 6.00 | 0.03 | 2.26 | 1.82 | 1.98 | 8.13 | 7.18 |
| Std.dev | 6.87 | 0.41 | 1.57 | 5.55 | 1.33 | 4.28 | 4.24 |
| Coef.Var | 91.89 | 244.13 | 67.05 | 151.68 | 67.05 | 46.28 | 58.00 |

* tabular ground ice samples are marked with an asterisk.

### 4.4. DOM Fractions

The Bulk fluorescent dissolved organic matter (fDOM) amounts calculated as a sum of the Raman-normalized fluorescence intensities for the fluorophores identified on EEM spectra (hDOM-1, hDOM-2, tyrDOM, and trpDOM) are reported in Table 1.

The characteristic EEM spectra, representative of highly variable patterns of the identified fluorophores, are shown in Figure 4. The measured EEM peak intensities (in Raman units, RU) and DOC-normalized fDOM value (RU/mg × L$^{-1}$C) are given in Table 5 and Figure 5a. DOC-normalized integral intensities of the humic-like (hDOM) and protein-like (pDOM) species of DOM are illustrated by Figure 5b. Maximum fDOM level was found in TGI sample #8 (4.91 RU), the minimum fDOM content was measured in ice-rich bubbly ice-wedge sample #12 (0.24 RU), which is characterized ultimately by tyrDOM. Median fDOM concentration was 1.604 RU with a standard deviation of 1.237 RU. The fDOM composition was mainly formed by hDOM (hDOM-1+hDOM-2) fraction making up >88% of the reported median content. hDOM-1 represents the higher molecular weight (HMW) humic-like DOM, while hDOM-2, discerned by a shorter excitation wavelength, represents a fraction of lower MW hDOM. TyrDOM, the most abundant protein-like DOM (pDOM) fluorophore, was detected in all the 16 samples with a median value of 0.29 RU (Figure 5a). In contrast another pDOM constituent-trpDOM was above the detection limit only in four TGI samples: (6, #8, #32, #33). In contrast trpDOM, another pDOM constituent, was above the detection limit only in four TGI samples: (6, #8, #32, #33). Distribution of the DOC (carbon)-normalized hDOM and pDOM (TrpDOM+TyrDOM) values shown in Figure 6b indicates a higher proportion of pDOM in the net fDOM for TGI samples compared to ice wedge. The diagnostic indices of DOM origin (FI and BIX) showed very minor variations, except for ice-wedge sample #12 in which FI = 2 and BIX = 0.8 are indicative of the ultimately autochtonous (microbial) origin of DOM, made up by exclusively TrpDOM. The DOC concentration in the sample#12 (5.49 mg/L) is quantitatively characteristic of dissolved carbon pool not affected by the contribution from generally predominant hDOM, responsible for variations in the ground ice DOC. Significant correlation of fDOM intensities to DOC concentrations (r = 0.8; $p$ = 0.0002) (Figure 6a) approved the adequacy of the utilized methodology of the fluorescence peaks quantification.

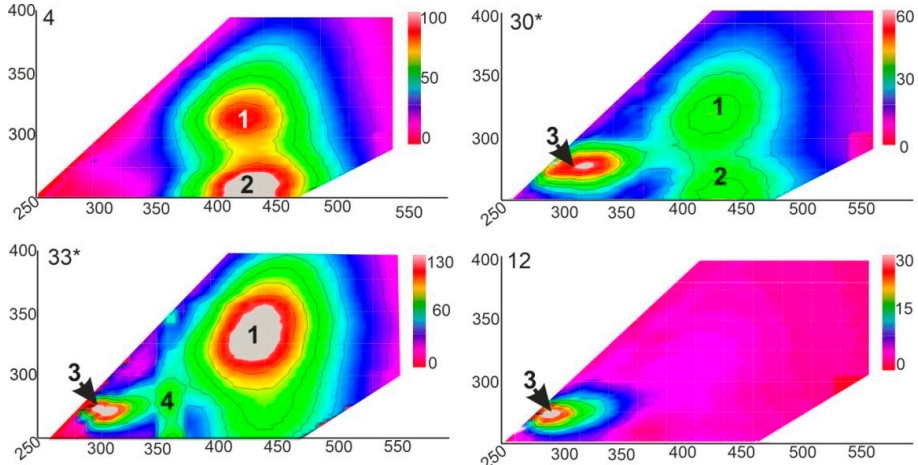

**Figure 4.** The representative excitation-emission (EEM) spectra of DOM, sample numbers are indicated in the left upper part of each spectrum. The peak marks are 1—hDOM-1; 2—hDOM-2; 3—tyrDOM; 4—trpDOM. * tabular ground ice samples are marked with an asterisk.

**Table 5.** Fraction composition of dissolved organic matter (DOM).

| Sample # | fDOM, RU | hDOM-1, RU | hDOM-2, RU | TyrDOM, RU | TrpDOM, RU | fDOM, RU/mg × L$^{-1}$C | BIX | FI |
|---|---|---|---|---|---|---|---|---|
| 1 | 3.38 | 1.70 | 1.26 | 0.42 | 0.00 | 10.55 | 0.4 | 1.5 |
| 2 | 3.06 | 1.44 | 1.16 | 0.46 | 0.00 | 10.52 | 0.5 | 1.7 |
| 3 | 2.72 | 1.23 | 1.11 | 0.38 | 0.00 | 13.46 | 0.6 | 1.6 |
| 4 | 0.87 | 0.50 | 0.32 | 0.04 | 0.00 | 6.59 | 0.6 | 1.6 |
| 10 | 0.96 | 0.28 | 0.30 | 0.39 | 0.00 | 6.22 | 0.6 | 1.6 |
| 11 | 1.68 | 0.74 | 0.67 | 0.27 | 0.00 | 7.34 | 0.5 | 1.6 |
| 12 | 0.24 | 0.00 | 0.00 | 0.24 | 0.00 | 5.49 | 0.8 | 2.0 |
| 27 | 0.80 | 0.27 | 0.28 | 0.26 | 0.00 | 4.89 | 0.5 | 1.5 |
| 28 | 2.54 | 1.21 | 1.21 | 0.12 | 0.00 | 10.98 | 0.5 | 1.6 |
| 29 | 1.53 | 0.67 | 0.67 | 0.18 | 0.00 | 8.20 | 0.5 | 1.7 |
| 30 | 0.39 | 0.09 | 0.16 | 0.14 | 0.00 | 8.06 | 0.6 | 1.6 |
| 06 * | 2.09 | 0.65 | 0.39 | 0.67 | 0.38 | 13.34 | 0.6 | 1.5 |
| 08 * | 4.91 | 2.32 | 1.08 | 0.96 | 0.54 | 21.42 | 0.8 | 1.6 |
| 32 * | 1.45 | 0.47 | 0.45 | 0.31 | 0.23 | 6.58 | 0.6 | 1.5 |
| 33 * | 1.06 | 0.31 | 0.28 | 0.25 | 0.23 | 4.51 | 0.6 | 1.6 |
| 34 * | 1.80 | 0.59 | 0.51 | 0.70 | 0.00 | 9.65 | 0.7 | 1.6 |
| Median value | 1.60 | 0.62 | 0.48 | 0.29 | 0.23 | 8.13 | 0.59 | 1.58 |
| Std.dev. | 1.23 | 0.64 | 0.42 | 0.24 | 0.20 | 4.27 | 0.10 | 0.12 |
| Coff.Var. | 67.10 | 81.83 | 67.95 | 66.57 | 73.52 | 46.28 | 18.39 | 7.83 |

* tabular ground ice samples are marked with an asterisk.

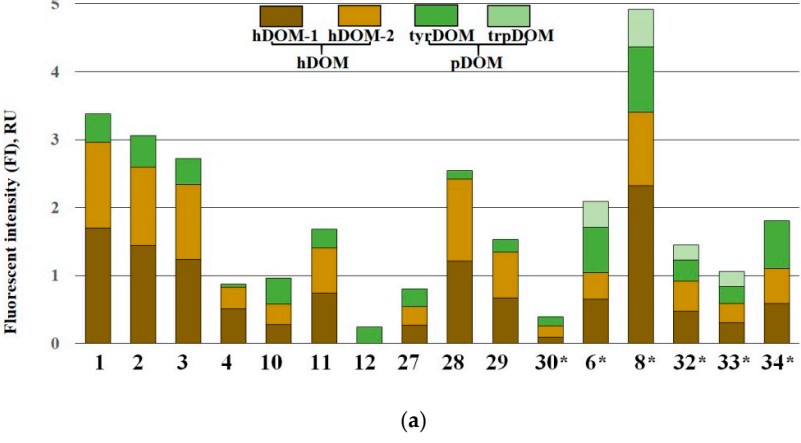

(**a**)

**Figure 5.** *Cont.*

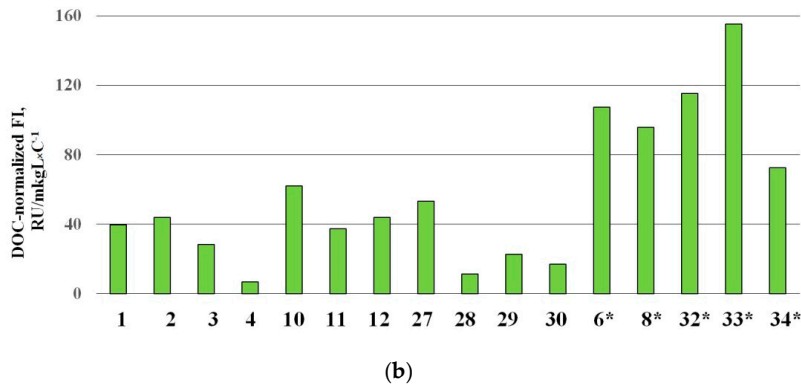

(b)

**Figure 5.** Distribution of fluorescent DOM (fDOM). (**a**) The molecular fractions of fDOM; (**b**) Distribution of the integral humic-like DOM (hDOM) and protein-like DOM (pDOM) values normalized to Dissolved organic carbon (DOC) concentration (Carbon-normalized values). * tabular ground ice (TGI) samples are marked with an asterisk.

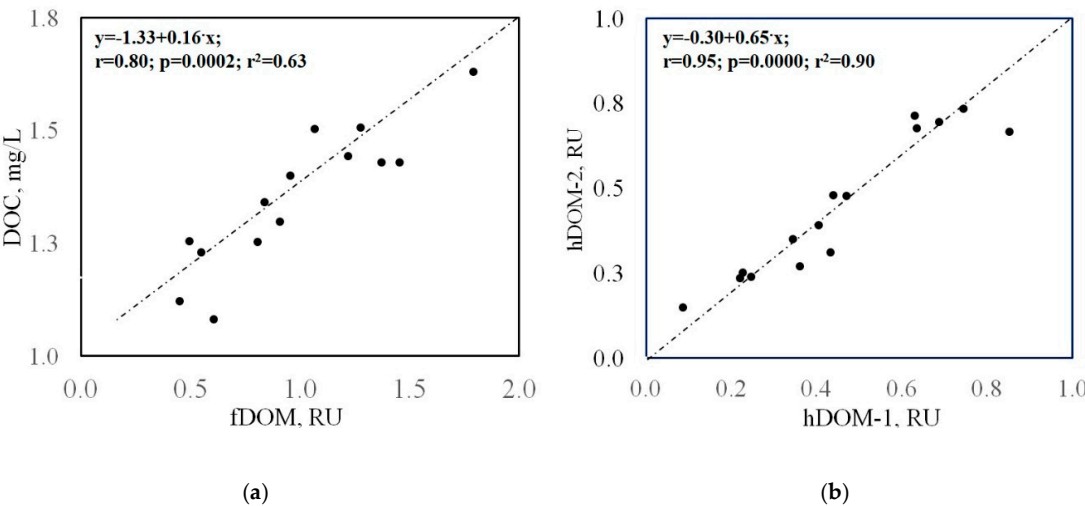

(a)　　　　　　　　　　　　　　　　　　　　　　(b)

**Figure 6.** (**a**) Bulk fluorescent dissolved organic matter (fDOM,) intensity against the DOC concentrations; (**b**) Humic like fluorophores (hDOM-1 and hDOM-2) relationship. The values were box-cox transformed to fit data to normal distribution.

## 5. Discussion

### 5.1. Methane Genesis, Storage, and Cycling

Based on the assumption of the similar conditions for the ground ice layers formation within the studied section we may suppose a common initial source of methane. The headspace diagram [30] allows tracking of the molecular fractionation in the gas mixture due to methane post-production alterations (Figure 4). The ground ice samples make up a four groups on the diagram, in which the group M (methane-enriched) is supposed to represent the less fractionated (methane depleted) HCG composition corresponding to predominantly methanogenic conditions. The group F2 shows the most fractionated HCG composition which can be related to the the methane oxidation environments The majority of ice-wedge ice shows the fractionated gas composition, probably due to aerobic methanotrophy. The TGI (Figure 7), samples #8, and #30, in contrast, have never been exposed to aerobic conditions, so they likely have retained the intact gas composition. The samples of methane-enriched ground ice units form two distinct pairs (#33, #8 and #6, #30) on both headspace and Bernard [39] plots (Figure 7a,b), demonstrating the correlated trends of HG molecular composition

and isotope signatures of $CH_4$. The synchronized shift in $\delta^{13}C$ and $C_1/C_2+C_3$ values may reflect the signatures of slightly oxidized methane in the samples #6 and #30 in comparison to those of the samples #8 and #33. On the CD diagram [15] which has long become a classic of methanogenesis pathway diagnostic, the two-by-two positioning of the methane-enriched samples #8 and #33 on headspace and Bernard plots remains (Figure 8). Isotope signatures ($\delta^{13}C$ and $\delta D$) of TGI samples #8 and #33 do not fall in the range of the conventional methane generation pathways summarized in the CD diagram. Based on the available data on hydrogen isotope composition of the ground ice thaw water we have attempted to characterize methanogenic conditions in the methane-enriched samples. Our interpretation is relevant as the TGI-extracted water bears the isotope signatures of the original methane formation [31,40]. Hydrogene isotope fractionation in the sulfate-free aquatic environments was reported to be directly linked to isotope composition of the coexisting water, but not affected by the kinetic isotope effect (KIE) coupled to methanogenesis as well as by methanogenic pathways relative proportion [31]. Using the Equation (1) and the available data on $\delta D$-$H_2O$ values of TGI samples #8 and #33 (−16 and −173 ‰, respectively), we obtain the predicted $\delta D$-$CH_4$ values of −39 and −400‰, respectively, that can be compared to the measurements (−39 and −397‰). However, the same calculation applied for the ice-wedge sample (#30) revealed the inconsistency between the predicted and the measured values of $\delta D$-$CH_4$, indicating the exogenous source of $CH_4$ enrichment. It is well known that ice wedge, in contrast to TGI, provides inappropriate conditions for methane production and accumulation [12]. $\alpha D$ values of the samples #8 and #33 obtained using the Equation (2) were 1.374 and 1.372, respectively, which confirms the genetic integrity of the corresponding methanogenic settings, at least in terms of hydrogen isotope fractionation. Relatively low values of $CH_4$-$\delta^{13}C$ (<−80‰, PDB) in these samples are indicative of carbonate reduction, which is significant for the net microbial methane generation. At the same time, deuterium (D) depletion of $CH_4$ ($\delta D < -350‰$), usually attributed to acetoclastic methanogenesis, may also be caused by the uptake of D-depleted water from carbonate reduction [41]. Relatively light isotope signatures of $CH_4$ from TGI samples #8 and #33 are probably linked to the enhanced KIE of carbonate reduction, due to low-temperature conditions. Based on the available data on western Yamal ground ice, we may assume a $\delta^{13}C$ $CH_4$ value of −70‰ as "background" for methane-rich TGI [12]. The simple calculation shows that the measured value of $\delta^{13}C$ (−84‰) could appear as result of equivalent mixing of the "background" $^{13}C$-relatively enriched methane with the $^{13}C$-depleted constituent of $\delta^{13}C$ value as low as −98‰. Such a negative signature of the proposed endmember suggests the strong fractionation, probably related to low temperature.

Carbonate reduction operated by anaerobic archaea under cryogenic conditions has been proved to exist in the permafrost environments [41,42]. The resulting methane is considered to be the cause of the drastic depletion in $^{13}C$, due to the KIE of enzymatic $CO_2$ fixation, enhanced at subzero temperatures [14]. The cryogenic methane formation in mid-Pleistocene epicryogenic units of East Siberia showed the values of $CH_4$-$\delta^{13}C$ as low as −99‰ [41]. The lower isotope fractionation characterized the higher temperature subzero methanogenesis was recorded in Holocene-Pleistocene deposits of submarine permafrost located within the Laptev Sea shelf. The related value of $CH_4$ $\delta^{13}C$ was as high as −72.2‰ PDB [42]. Considerable depletion in deuterium ($dD \approx -479‰$) has been reported for $CH_4$ of Siberian permafrost [43]. Kinetic measurements using the $^{14}C$–labeled substrates allowed the quantifying of the detectable but extremely slow rates of a cryogenic $CH_4$ production by the permafrost-inhabiting methanogenic archaea [41,44]. The microbial turnover of carbon and sulfur species under frozen conditions has been concluded from the isotopic studies of the Western Yamal TGI layers [45].

The sites of the local $CH_4$ accumulations were found at different sampling depths of the exposure (7.96–4.9 m) and at a considerable horizontal distance from each other. Such a large variation in methane content, as well as the relative abundance of methane in tabular ground ice as compared to ice wedges, is consistent with the previously published data on the ground ice of Yamal Peninsula [12,40,46].

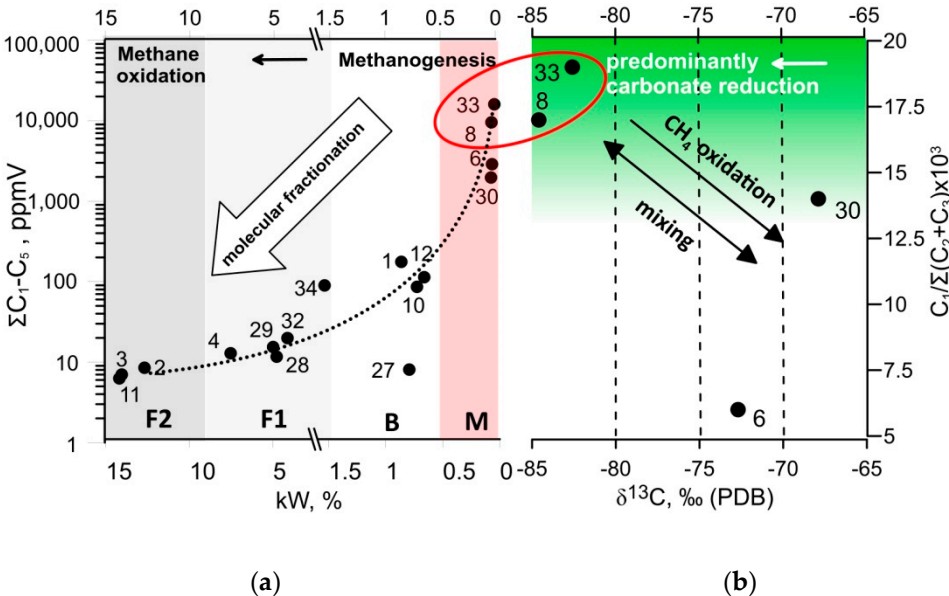

(a)　　　　　　　　　　　　　　(b)

**Figure 7.** (**a**) Headspace diagram of $C_1$–$C_5$ HG (modified after [30]), the letters below indicate the grouping of the ground ice samples in terms of methane cycling based on molecular composition of the hydrocarbon gases (HCG): M—methane enriched; B—background; F1 and F2—fractionated (methane depleted); (**b**) Bernard diagram of the molecular ratio of $C_1$/$C_2$+$C_3$ against $\delta^{13}$C $CH_4$ (modified after [39]). Red ellipse indicates the TGI samples with similar molecular of $C_1$-$C_5$ HG composition of and $\delta^{13}$C $CH_{4\,v}$ values

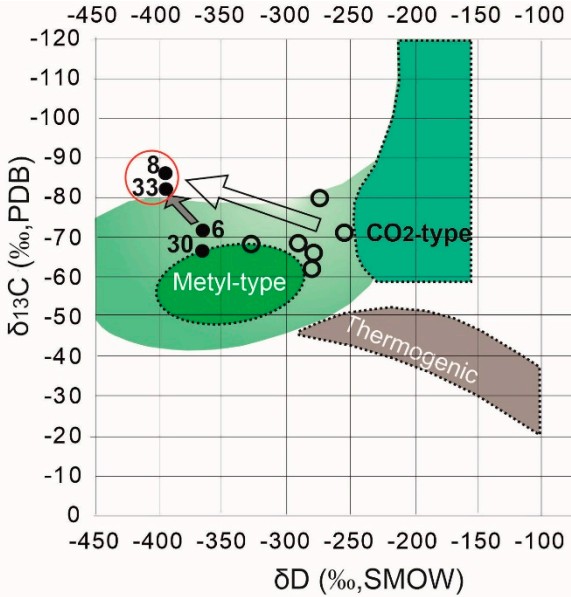

**Figure 8.** CD diagram for diagnostics of $CH_4$ generation pathway (after [15]); empty circles indicate the reference (published) data from Western Yamal nearshore permafrost landscapes [12]; the big transparent arrow marks the variation of $CH_4$ isotope signatures between the western Yamal (reference) and the Central Yamal (target) ground ice deposits; grey arrow points at variation of $CH_4$ isotope signatures between the methane-rich ground ice samples of the central Yamal thermocirque (target area). Red oval outline the TGI samples with anomalous methane content.

Thus, the sampling sites # 34 and # 33 (Figure 3) are located within the thermocirque wall as close to each other as 1 m, but the corresponding TGI samples contain an amount of methane as different as 200 times. Such a local character of methane accumulation within the frozen deposits can be explained

by in situ methanogenesis. For the sampled deposits it suggests the slow rate of methane production for at least 37,000 years (according to the results of peat dating [24]) at the lack of an opportunity for the produced methane to naturally dissipate by molecular diffusion [14].

Considering the relevance of the sub-zero methanogenesis proposed phenomena, the reported integrity of methanogenic conditions for the TGI samples #8 and #33 may be attributed to survival conditions of archeal population, hypothetically responsible for the observed methane enrichment.

Moreover, the TGI samples #8 and #33 are dramatically different in physical properties and composition. Thus, the sample #8 is characterized by 80% of the soil fraction, in contrast to the ice-rich TGI sample #33 that is almost totally lacking in the soil inclusions, but having twice as much air bubbles as the sample #8.

Concerning the proposed significance of subzero methanogenesis, there should be no significant limitations from the mentioned compositional divergence on the activity of methanogensic community in the corresponding environments.

Published data on of the Western Yamal ground ice indicate $CH_4$ $\delta^{13}C$ values ranging from $-62$ to $-74$‰ with an average of $-68$‰ and $\delta D$ variation from $-259$ to $-330$‰ [40]. The collected data from the Central Yamal ground ice deposits show the values of $CH4$ $\delta13C$ varying from $-72.6$ to $-82.4$‰ with an average of $-77.0$‰ and D values ranging from $-369$ to $-397$‰ with a mean value of $-385$‰ (Figure 8). Perhaps there is a trend of decrease in both $\delta^{13}C$ and $\delta D$ values of the ground ice-derived methane, heading from Northern to Central Yamal. However, the available data are insufficient for reliable interpretation of the large-scale variations.

## 5.2. Fluorescent Dissolved Organic Matter (fDOM) Composition and Labile DOM Contribution to Biogeochemical Cycling

Fluorescent dissolved organic matter (fDOM) composition is linked to the biochemical availability of the ground-ice derived relic organic matter. The observed fDOM to DOC correlation (Figure 6a) reveals the fDOM predominance in the not-measurable net DOM of the ground ice samples, suggesting a minor contribution from non-fluorescent dissolved organic compounds. Significant correlation between humic-like DOM (hDOM1 and hDOM2) intensities suggests a common allochthonous source of these constituents (Figure 6b) not related to the autochthonous source, marked by pDOM. The appearance of the trpDOM flourophore, presumably indicating the highly degraded proteins, peptides and/or free amino acids (AA) [47,48], is likely more common for TGI. Thus, it was recorded in four of five TGI samples, while it was not detected in any of the 11 ice-wedge samples [49,50]. Accumulation of highly reactive LMW intermediates (AA and $NH_4^+$) in TGI units suggests that the carbon cycle operation was not limited by the available nitrogen in the source soil environment before the freezing. It may point to relatively poor higher plants and vegetation which usually compete for available nitrogen with soil microbiota. The correlation between the values of the source pDOM and the resulting DIN, is likely a feature of the TGI samples (Figure 9b). Ice wedges appear to be more depleted in available nitrogen species (AA and DIN) than TGI, probably due to an uptake by aerobic microbiota, functioning under atmospheric conditions. Large protein molecules are presumably represented by trpDOM and could be released by microbial biomass regeneration. Another factor affecting the ice-wedge environments is the snowmelt water mixing which may reduce the concentrations of metabolically active solutes.

## 5.3. Biogeochemical Environments and Their Potential for the GHG Liberation upon Thawing

Highly heterogeneous biogeochemical settings made up of the ground ice layers are exposed to periodical thawing and can be detected by the thermocirque wall sampling. The resulting mudflows are mixed up together and rapidly flow down the slope into the local catchments at the foot of the thermocirque [18,51]. Export of humic compounds (evaluated as CDOM) by thawing TGI layers into the adjacent water catchment has been documented for the Central Yamal landscapes [18]. However, the vast carbon pool associated with allochtonous DOM (CDOM, hDOM), is refractory to microbial mineralization, suggesting its predominant accumulation over decomposition.

The slow multistep processing of the accumulated OM, finally yielding the substrates consumable by methanogens, is responsible for a pronounced lag phase setting up a special time scale of methanogenesis and methane emission. While the thaw-induced mobilization of the ground ice-stored $CH_4$ and coinciding $CO_2$ production by aerobic respiration of the released labile DOM can be observed and evaluated within a common short-term time scale [52]. DOC biodegradability is well known for being stimulated by high concentrations of DIN, indicating the intimate relations between carbon and nitrogen cycles [53]. Considering the rough qualitative evaluation of the encountered biogeochemical environments by the potential for a quick/short-term GHG release upon ground ice thawing, we have reduced the available dataset to the three main factors: dissolved $CH_4$ content (ng/g), pDOM fluorescence intensity (RU) and DIN concentration (mg/L). The quantity of the biodegradable DOC can only be defined by the in vitro incubation tests [7,52]. In this work, we use pDOM fluorescent intensity as an indicator of the relative labile DOM pool size. Values of carbon-normalized pDOM fluorescent intensity (RU/mkg $\times$ L$^{-1}$ $\times$ C) are characteristic of DOM quality. The higher quality of TGI-derived DOM is provided by the better preservation conditions (Figure 5b). The simultaneous monitoring of the BDOC content and the fDOM composition in the inoculated meltwater samples under the recommended incubation conditions will allow for a better understanding of the pDOM composition dynamics in relation to the BDOC conversion to $CO_2$ [7].

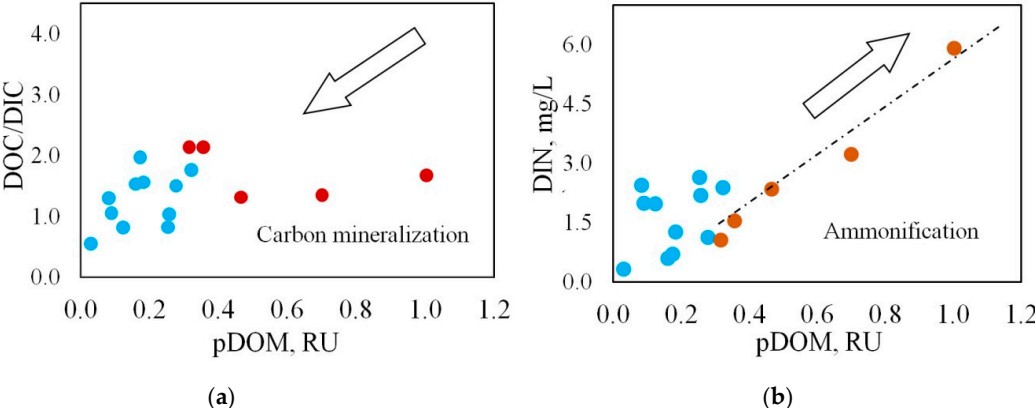

(**a**)　　　　　　　　　　　　　　　　　　　　　(**b**)

**Figure 9.** Relationship between protein-like DOM (pDOM) fluorescence intensity and (**a**) DIC/DOC ratio; (**b**) dissolved inorganic nitrogen DIN; arrows indicate the major trends in biogeochemical cycling: DOC mineralization (carbon cycle) and ammonification (nitrogen cycle). Red circles—TGI, blue circles—ice wedges.

The histogram visualizes the natural variation in the selected key factors of GHG emission (Figure 10). We assume that the TGI is a major contributor of $CH_4$, as well as of the dissolved nutrients and labile DOM released upon thaw. The type of biogeochemical environment represented by the TGI sample #8 is supposed to be of particular importance as a potential provider of the intensive carbon cycle inventory. We may speculate that the abundance of the fine-grained soil fraction in the resulting meltwater stream would reduce the methane migration escape to the atmosphere [54], while ice-rich environments with a sufficient amount of free gas inclusions would likely promote fast methane diffusion and volatilization. The setting, represented by TGI sample #33 bears the largest pull of $CH_4$ as well as the favorable conditions of methane loss upon thaw. The bubbly ice-wedge ice #4 shows the lowest amount of the selected compounds, thus indicating the example of the biogeochemical environment with relatively low potential for GHG release upon thaw.

The utilized approach could be a simple but informative screening tool for qualitative evaluation of the heterogeneity of the cryogenic landscape concerning GHG emission as well as for comparison of the different permafrost settings. The involvement of the BDOC pool quantification in the typical ground ice environments is a necessary step for future research in this field.

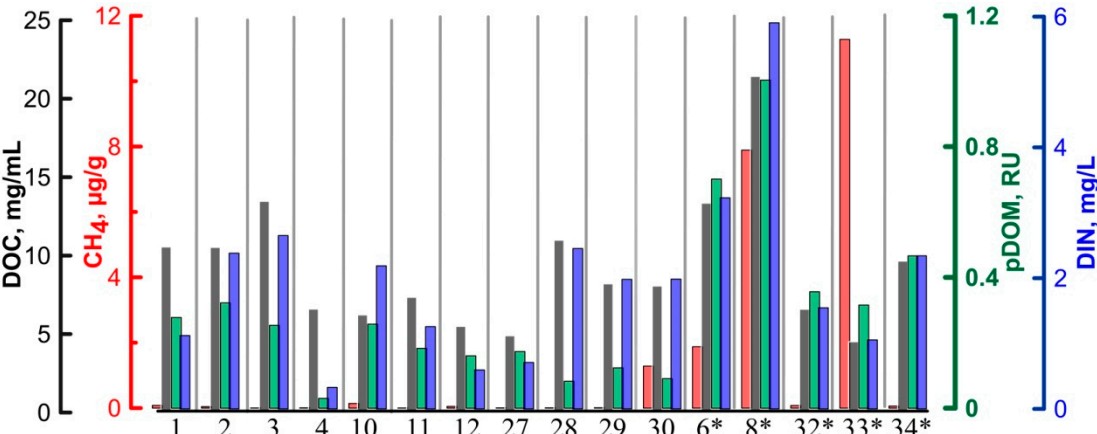

**Figure 10.** Distribution of the selected key biogeochemical factors in the ground ice samples. * -tabular ground ice (TGI) samples are marked with an astersisk.

## 6. Conclusions

Thermocirque exposure, located in the central part of the Yamal Peninsula, was sampled for the biogeochemical study. Laboratory analysis of the ground ice samples revealed a series of highly divergent biogeochemical environments with the patterns of the key parameter distribution mainly governed by the ground ice formation conditions.

The isotopic signatures of methane ($\delta^{13}C$ and $\delta D$, ‰), as well as the values of hydrogen fractionation factor ($\alpha D$) between methane and methane-formation water, suggest the strikingly similar methanogenic conditions in the methane-rich tabular ground ice units, irrespective of the very different basic properties shown by the corresponding ground ice samples. The enormously light isotope composition of the anomalous $CH_4$ ($\delta^{13}C < -80$‰, $\delta D < -390$‰) can be explained by mixing of the "background" methane pool with isotopically light end member (depleted in both $\delta^{13}C$ and $\delta D$), produced by carbonate reduction under cryogenic conditions.

Available data indicate a clear negative trend in the distribution of both carbon and hydrogen isotope signatures of ground ice-derived methane, heading from Northern to Central Yamal. This could be related either to some pronounced variations in methane generation physical conditions or to some geographic patterns of isotope composition in precursor substrates.

The dissolved organic matter in the analyzed ground ice samples is mainly formed (>88%) by biochemically refractory allochtonous humic acids. Using the protein-like DOM fluorophores as indicators of the labile autochtonous OM, we observed the features of preferential involvement of the associated carbon pool in microbial mineralization. This trend is expressed to various degrees for both: TGI samples, preserving the oxygen-depleted soil environments, and ice-wedge ice, subjected to oxygenated conditions during formation. Considering the nitrogen cycle, we detected the features of ammonification for the TGI samples. Ice wedges, in contrast, revealed relative depletion in dissolved nitrogen species as well as in pDOM composition. This may be related to the enhanced demand in available nitrogen by the aerobic microbiota, operating in the ice-wedge environments. Biogeochemical cycling in ice wedges should also be affected by mixing with atmospheric water.

Distribution of the selected biogeochemical parameters enables the ranking of the representative ground ice environments with implication to GHG emission upon thawing. Within the sampled section, the TGI layers are of the higher potential for GHG emission than the ice wedges, owing to the higher content of the stored $CH_4$, a better quality of DOM, and higher nutrient loadings. Ice wedges express a high variation in DOM composition and lability. In this work, we have tried a simple screening tool for qualitative evaluation of the cryogenic landscape heterogeneity for GHG emission as well as for comparison of different permafrost settings. The involvement of the quantitative tools for biodegradable DOC determination, coupled with simultaneous DOM composition, and the selected

carbon and nitrogen cycles intermediate monitoring have been proposed as a further development of this research.

**Author Contributions:** Conceptualization, P.B.S. and A.A.P.; methodology, P.B.S., S.A.M., E.V.S., and A.I.K.; P.B.S., A.A.P. writing—original draft preparation, M.O.L., I.D.S.; writing—review and editing, B.G.V.; supervision. Please turn to the for the term explanation All authors have read and agreed to the published version of the manuscript.

**Funding:** Field and laboratory work of this research was funded by RFBR according to the research project 18-05-60004, analytical work was funded by RFBR according to the research project 18-05-60080 иand partial support within the framework of the State assignment on the topic "Earth Cryosphere Change under the Influence of Natural Factors and Technogenesis" NIR AAAA-A16-116032810095-6, PP 55 Arctic. Stable isotope studies in this research were funded by RSF 19-17-00226.

**Acknowledgments:** The authors would like to thank Victor Bogin (VNIIOkeangeologia) for the great assistance in the ground ice sample processing, Alexey Krylov (VNIIOkeangeologia) for valuable advice on isotope geochemistry, anonymous reviewers for the detailded reports allowed to significantly improve the manuscript.

**Conflicts of Interest:** The authors declare no conflict of interest. The funders had no role in the design of the study; in the collection, analyses, or interpretation of data; in the writing of the manuscript, or in the decision to publish the results.

## Abbreviations

| | |
|---|---|
| DOC | Dissolved organic carbon |
| DOM | Dissolved organic matter |
| GHG | Greenhouse gas |
| OM | Organic matter |
| BDOC | Biodegradable dissolved organic carbon |
| TGI | Tabular ground ice |
| CDOM | Colored dissolved organic matter |
| ALD | Active layer depth |
| MIS | Marine isotope stage |
| HG | Light hydrocarbon gases |
| GC | gas chromatograph |
| DIC | Dissolved inorganic carbon |
| GF/F | glass microfiber syringe filter |
| IC | ion chromatograph |
| DIN | Dissolved inorganic nitrogen |
| EEM | Excitation Emission Matrix fluorimetry |
| FI | fluorescent index |
| BIX | biological index |
| fDOM | fluorescent dissolved organic matter |
| hDOM | humic-like dissolved organic matter |
| fDOM | fluorescent dissolved organic matter |
| pDOM | protein-like dissolved organic matter |
| TrpDOM | tryptophan-like dissolved organic matter |
| TyrDOM | tyrosine-like |
| KIE | kinetic isotope effect |
| LMW | low molecular weight |
| IRMS | isotopic-ratio mass spectrometry |

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
