# Peer review of "Methane and Dissolved Organic Matter in the Ground Ice Samples from Central Yamal: Implications to Biogeochemical Cycling and Greenhouse Gas Emission"

_geosciences, doi:10.3390/geosciences10110450_

Round 1
Reviewer 1 Report
I feel that authors have tried their best to address reviewers' comments and concerns in their revision. a. This is a stronger manuscript than the one first submitted to this journal.
 Some corrections and few suggestions in the attached PDF. I recommend this second revision for publication.

Author Response
COVER LETTER
1.We have changed the title: “Methane and dissolved organic matter composition of ground ice in Central Yamal: Implications to biogeochemical cycling and greenhouse gas emission”
into
- “Methane and dissolved organic matter in the ground ice samples from Central Yamal: Implications to biogeochemical cycling and greenhouse gas emission”
Because “methane composition” may sound confusing
We have transferred the formula from Discussion in to the Methods section
- The Reference list has been reorganized in accordance with the text rearrangement
Thank you for attention to our manuscript and the valuable comments!
Line 72 It is
but it is not well understood…
Line 248 peat leanse
Peat lense
Line 251 THE minimum
The minimum
Line 254 that is rich enriched in both
enriched in…
Line 261 The bulk
The bulk
Line 264 (is) are reported in Table 1
The Bulk fluorescent dissolved organic matter (fDOM) amounts calculated as a sum of the Raman-normalized fluorescence intensities for the fluorophores identified on EEM spectra (hDOM-1, hDOM-2, tyrDOM, and trpDOM) are reported in Table 1
Line 275 by Figure 5b
In Figure 5b
Line 275 The maximum
The maximum
Line 276 represented characterized?
Represented characterized ultimately by TyrDOM
Line 282 In contrast another pDOM constituent - trpDOM was above the detection limit only in 4 TGI samples: (6, #8, #32, #33)
In contrast, the trpDOM, another pDOM constituent was above the detection limit only in 4 TGI samples: (6, #8, #32, #33)
Line 291-295 Significant correlation of fDOM intensities to DOC concentrations (r = 0,8; p = 0,0002) (Figure 6a) was the important issue, confirming the adequacy of the utilized DOM fractions measurement based on fluorescence peak identification and quantification methodology – Please can you better rephrase?
Significant correlation of fDOM intensities to DOC concentrations (r = 0,8; p = 0,0002) (Figure 6a ) approved the adequacy of the utilized methodology of the fluorescence peaks quantification.
Line 311 indicates shows
…shows
Line 318 ,
,
Line 319 elevated enriched
Line 320 previously noted (#8 and #33)
On the CD diagram [15], that have long become a classic of methanogenesis pathway diagnostic, the two-by-two positioning of the methane-enriched samples #8 and #33 on headspace and Bernard plots remains (Figure 8a).
Line 322 Involving Based on
Based on the available data
Line 324 Our interpretation is relevant if as ?
Our interpretation is relevant as the TGI extracted water…
Line 332 …the above equation and available data on δD-H2O values of TGI samples #8 and #33 (-167‰ and -173 ‰, respectively), we obtain the predicted δD-CH4 values of -396‰ and -400‰, respectively equal that can be compared to the measured (-394‰ and -397‰).
…the above equation and available data on δD-H2O values of TGI samples #8 and #33 (-167‰ and -173 ‰, respectively), we obtain the predicted δD-CH4 values of -396‰ and -400‰, respectively that can be compared to the measured (-394‰ and -397‰).
Line 333 However the similar At the same time, a similar calculation undertaken for applied to the ice-wedge sample (#30) revealed the inconsistency between the predicted and the measured values of δD-CH4, indicating the exogenous source of CH4 enrichment
However, the similar calculation applied to the ice-wedge sample (#30) revealed the inconsistency between the predicted and the measured values of δD-CH4, indicating the exogenous source of CH4 enrichment
Lines 335-336 It is well known that ice wedge, in contrast to TGI, provides the inappropriate conditions conditions inappropriate for methane production and accumulation
It is well known that ice wedge, in contrast to TGI, provides the inappropriate conditions for methane production and accumulation, the reference added !!!!!
Line 338. The obtained αD values was constitute 1,374 and 1,372 for samples of #8 and #33, respectively, confirming the genetic integrity of the corresponding methanogenic settings, at least, in terms of hydrogen isotope fractionation (Figure 8 b)
The obtained αD values of the samples # 8 and #33, were 1,374 and 1,372 respectively, which is confirming the genetic integrity of the corresponding methanogenic settings, at least, in terms of hydrogen isotope fractionation (Figure 8 b)
Line 341. carbonate reduction significance which is significant for the net microbial methane generation
carbonate reduction, is significant for the net microbial methane generation
Line 343 of D-depleted water from for carbonate reduction
of D-depleted water from carbonate reduction
Line 352 The resulting methane is considered to be the cause of is reported for drastic depletion in 13C, due to the KIE of enzymatic CO2 fixation, enhanced as a function of slow reaction rate at a subzero temperature [14]
The resulting methane is considered to be the cause of is reported for drastic depletion in 13C, due to the KIE of enzymatic CO2 fixation, enhanced at a subzero temperature [14]
Line 354 The proposed significance of cryogenic methane formation was shown to display showed the values of CH4-δ13C values as low as -99‰ in mid-Pleistocene epicryogenic units of East Siberia [41]
The cryogenic methane formation in mid-Pleistocene epicryogenic units of East Siberia showed the values of CH4-δ13C as low as-99‰
Lines 368-369 Thus, sampling sites #34and #33 (Figure 3) are located less than a meter apart within the thermocirque wall, but the corresponding TGI samples contain an amount of methane about 200 times different. Can u better rephrase?
Thus, the sampling sites # 34 and #33 (Figure 3) are located within the thermocirque wall as close to each other as 1 m, but the corresponding TGI samples contain an amount of methane as different as 200 times
Lines 374-376 I Considering the relevance n the view of subzero methanogenesis poroposed proposed significance, the reported integrity of methanogenic conditions for the TGI samples #8 and #33 may be attributed to survival conditions of 375 archeal population, hypothetically responsible for the observed methane enrichment 376
Considering the relevance of the sub zero methanogenesis proposed phenomena, the reported integrity of methanogenic conditions for the TGI samples #8 and #33 may be attributed to survival conditions of archeal population, hypothetically responsible for the observed methane enrichment
Lines 377-378. Thus, sample #8 characterized by is 80% of the soil fraction, in contrast to the ice-rich TGI sample #33 that is almost lacking in the soil inclusions, but having twice as much of air bubbles of the sample #8
Thus, the sample #8 is characterized by 80% of the soil fraction, contrast to the ice-rich TGI sample #33 that is almost lacking in the soil inclusions, but having twice as much of air bubbles of the sample #8
.Lines 397-399 The collected data Our data on the from the Central Yamal ground ice deposits sposits show the values of CH4 δ13C varying from -72,6 to -82,4‰ with an average of -77,0‰ and D values ranging from -369 to -397‰ with a mean value?of -385‰ (Figure 8)
The collected data from the Central Yamal ground ice deposits show the values of CH4 δ13C varying from -72,6 to -82,4‰ with an average of -77,0‰ and D values ranging from -369 to -397‰ with a mean value of -385‰ (Figure 8)
Lines 409-411. Significant correlation of the high molecular weight humic-like DOM (hDOM1 and hDOM2) intensities suggests a the common allochthonous source of these constituents (Figure 6b) not related to the autochthonous 410 source, marked by pDOM.
Significant correlation of the high molecular weight humic-like DOM (hDOM1 and hDOM2) intensities suggests a common allochthonous source of these constituents (Figure 6b) not related to the autochthonous 410 source, marked by pDOM
Lines 412-420 The appearance of tyrDOM flourophore, indicative of poorly hydrolyzed proteins contribution [47,48], is likely to be more common for TGI since it has been recorded in 4 of 5 TGI samples, but in none of 11 ice-wedge samples
This fragment was substationally reworked and rephrased into:
The appearance of trpDOM flourophore, presumably indicating the highly degraded proteins, peptides or free amino acids (AA) [47,48], is likely more common for TGI. Thus, it was recorded in 4 of 5 TGI samples, while in all the 11 ice-wedge samples it was not detected. [49,50]. Accumulation of highly reactive LMW intermediates (AA and NH4+) in TGI suggest that the carbon cycle operation was not limited by the available nitrogen in the source soil environment before freezing. It may point on relatively poor higher plants vegetation which usually compete for available nitrogen with soil microbiota. The correlation between the values of the source pDOM and the resulting DIN, is likely a feature of the TGI samples (Figure 9b). Ice wedges are more depleted in available nitrogen species (AA and DIN) than TGI, probably due uptake by aerobic microbiota, which is functioning under atmospheric conditions. Large protein molecules are presumably represented by trpDOM, could be released by microbial biomass regeneration. Another factor affecting the ice-wedge environments is the snowmelt water mixing wich may reduce the solutes.concentrations.
Lines 436-438 Highly heterogeneous biogeochemical settings made up of ground ice layers are exposed to periodical thawing and detectable by thermocirque wall sampling.
Highly heterogeneous biogeochemical settings made up of ground ice layers are exposed to periodical thawing and can be detected by the thermocirque wall sampling
Lines 445-447. While thaw-induced mobilization of the ground ice-stored CH4 and coinciding CO2 production by aerobic respiration of the released labile DOM, are likely to make up an immediately detectable signal of permafrost carbon, reintroduced into the global biogeochemical cycle [52]. Please can you better rephrase?
While the thaw-induced mobilization of the ground ice-stored CH4 and CO2 production by aerobic respiration of the released labile DOM, can be observed and evaluated within a common short-term time scale. [52].
Line 462-463 We assume that the TGI is a more suuficient contributor of CH4, as well as of the dissolved nutrients and the labile DOM released upon thaw
We assume that the TGI is a major contributor of CH4, as well as of the dissolved nutrients and the labile DOM released upon thaw
Line 497 This trend in various degrees is expressed for both: TGI samples, preserving the oxygen-depleted soil environments
This trend to various degrees is expressed for both: TGI samples, preserving the oxygen-depleted soil environments
Line 499 In respet to considering the nitrogen cycle, we detected the features of ammonification for the TGI samples
Considering the nitrogen cycle, we detected the features of ammonification for the TGI samples
*I suggest to report all the calculation equations in the methods section and refer to them in the discussion
We have relocated all the equations for isotope calculation (lines 326, 337) into the corresponding paragraph of the Materials and Methods section as it has been suggested by Reviewer , except the calculation (Line 347) , which is just a simple arithmetic ratio .
The liner regression equation based on both laboratory and field data from natural sulfate-depleted freshwater environments was was used to evaluate the fractionation between δD-CH4 on δD-H2O in the ground ice samples [40]:
δD-CH4=0,675*(δD-H2O)-284‰ (p<0,0005) (1)
Hydrogen isotope fractionation factor between methane and methane formation water was calculated as [Whiticar, 1999] :
αD = (δDH2O + 103) / (δCH4 + 10)3). (2)

Reviewer 2 Report
Authors have addressed all my comments. I think the paper is now acceptable for publication. Minor editing (language) needed.
Author Response
COVER LETTER
1.We have changed the title: “Methane and dissolved organic matter composition of ground ice in Central Yamal: Implications to biogeochemical cycling and greenhouse gas emission”
into
- “Methane and dissolved organic matter in the ground ice samples from Central Yamal: Implications to biogeochemical cycling and greenhouse gas emission”
Because “methane composition” may sound confusing
We have transferred the formula from Discussion in to the Methods section
- The Reference list has been reorganized in accordance with the text rearrangement
Thank you for attention to our manuscript and the valuable comments!

This manuscript is a resubmission of an earlier submission. The following is a list of the peer review reports and author responses from that submission.
Round 1
Reviewer 1 Report
The paper deals with an interesting topic, the implications to biogeochemical cycling and greenhouse gas emission derived by the study of light hydrocarbon gases and dissolved organic matter in ground ice in Central Yamal. It is an important theme and the authors present interesting data and aspects but the manuscript suffers the lack of a clear structure and analytical results in are not always presented in an adequate way
In several cases an excessive amount of details and a not linear syntactical structure make the comprehension of the text quite difficult. A revision of the English is necessary. Please avoid generic expression: one can..., corresponding..
Still not clear where published data from literature stops and new results of the study starts
The introduction needs to be re-organized in order to better underline the goal of the study and report the state of research in the area. Several sentences need to be rephrased because not clear.
The study area description as well should be re-organized (e.g. lines 102 -105, lithologies description, should be move close to lines 115-116), and some statements require better explanations adopting a proper geological description and terminology (e.g. lines 126-127 state “TGI formed during the freezing of water-saturated sediments immediately after the sea bottom came to the surface”.
Results are interesting but concerns rise by the comparison of the geo-biochemical signature of samples and their position in the section:
Samples 8 and 33 seems to have a similar signature but they are located in different portions of the massive tabular ground ice; in addition samples 34 and 32, that are close to sample 33, show a different behavior (Figure 10). A deeper discussion should help to focus the results.
Sampling depth meters reported in figure 3 do not coincide with those reported in table 2. Table 2 reports depths from field level while figure2 and 3 report depths respect to sea level? (please add acronyms).
Please find in the attached PDF my comments and remarks.

Reviewer 2 Report
General comments:
The manuscript deals with the chemical and stable isotopic (carbon and hydrogen) composition of gases trapped in ice, and the conversion of organic matter into greenhouse gases. Implications for biogeochemical cycles and greenhouse gases emissions are discussed. The authors use a wide range of analytical techniques, resulting in a large dataset to support their discussion. The manuscript is interesting and adequate for publication in Geosciences. In general, I found the paper difficult to read and “digest”. I would recommend some revision of the language, notably the structure of the sentences, to make the paper easier to read. In addition, the authors use a large number of abbreviations (e.g., HDOM, pDOM, tyrDOM, hDOM) that contribute to make the reading more difficult. It might be reasonable to spell out the abbreviations in some parts of the text. I would also recommend a better separation of the Results from Discussions (see comment in Line below)
Specific comments:
Line 32: I would suggest to use “climate change” instead of “climate warming”, as the climate itself may become warmer or colder under the contemporary climate change trend. Please replace this in the entire manuscript. Alternatively use the term “global warming”.
Line 33: Replace “blocked by” with “locked in”. Why not use the term permafrost instead of “geo-cryological”?
Lines 42-43: I would prefer “aerobic” instead of “oxic” and “anaerobic” instead of “anoxic”.
Line 52: “By this time”...which time? Please clarify this in the sentence.
Line 73: Why is this a pilot work? It seems to be a “regular” work.
Line 75: Add “in the” before “modern carbon cycle”.
Lines 77-78: The objective of the paper is not clearly defined here. Please revise.
Line 89: “...passes close.” is a very imprecise term. Please revise that.
Lines 90-91: This sentence is not clear. Please revise.
Line 179: Replace DIC with DOC.
Lines 218-249: This paragraph mixes Results and Discussions. One example is the discussion of methane pathways (CO2 reduction vs. a acetyl fermentation).
Line 335: Replace “metanotrophy” with “methanotrophy”.
Line 357: “...over a geological time scale...” is too vague. Please define a time imagined for the process to be significant.
Lines 382-384: This last sentence sounds speculative. Provide a reference for it.
Line 461: Methane “conductivity” is probably inappropriate here. Perhaps “migration” would be more appropriated?
Fig. 1 – Please add a location map showing the study area in Russia (perhaps as an inset?). Geosciences is an international journal with a broad audience. In addition, this figure doesn’t show precisely the location of the study area shown in Figure 2.
Fig. 2 – Describe in the captions the meaning of the numbers on the photograph (altitude? Elevation to a reference level?). In addition, write that the samples are represented by the blue circles.
Figure 3 – Describe in the captions the meaning of bIII, aIII, and sIV (it may not be obvious for all potential readers).
Reviewer 3 Report
The manuscript describes the results of a project that focuses on Holocene deposits from an outcrop in the Yamal Peninsula, northwest Siberia, Russia. The authors have investigated stable carbon and hydrogen isotope compositions of methane as well as several biogeochemical parameters from samples obtained from two types of ground ice, i.e. tabular ground ice and ice wedges. The location of the study area and the nature of samples available for analysis are certainly unique. The availability of high-quality data, particularly in the context of green-house emissions and biogeochemical cycling in the polar regions would certainly be of significant interest to a broad range of climate scientists and biogeoscientists. In spite of the interesting hard-to-get data, the manuscript, however, contains several major issues that need to be addressed before it can be considered for publication.
There are significant issues with describing how the data were acquired (e.g. see the comment for lines 164-178), their availability (e.g. concentration of methane and C and H isotope data are not available for viewing, see the comment for line 220) and how they are discussed. See below for more detail. The manuscript needs a major revision that addresses these issues.
MAJOR ISSUES
First, a poorly written abstract
The abstract comprises a set of more or less disjointed statements about what was found during the project. The abstract needs to be re-written to make it clear why the authors have done the work (in other words, what knowledge gap(s) they are addressing), why the methodological approach they’ve chosen would be appropriate here, what specific data they’ve generated, and what is significant about their findings.
Second, a similar issue with the Introduction section
There is quite a bit of background about global warming in the polar regions, thawing permafrost, the role of green-house gases (CO2 and CH4) but very little about the specifics of the project, i.e. what research problem (in a form of hypothesis or research perhaps?) the authors are trying to address. Why is the region they chose for studying important? Is the research question they are trying to investigate of regional importance or are there global implications? On line 360, the authors mention “the hypothesis on subzero methanogenesis”. It would be very useful to have this mentioned in the Introduction section, so that the reader would have a better understanding of what the work described in this manuscript is trying to achieve.
Third, the Results section
The section should be used to show and describe the data that were acquired in this project. Instead, the authors include references, construct figures and then discuss their results based on data from other publications. If the authors maintain on this sort of approach, this section should be merged with the Discussion section.
Fourth, the Discussion section
When discussing methane genesis, storage and cycling, the authors use only 4 samples. What was the rationale for choosing only these samples? How what the interpretation of methane origin in these ice bodies change if more methane sample were investigated? Are these 4 samples representative of the whole group of samples obtained at the outcrop?
Fifth, the two sections on lines 385-431 and 432-472
It is not clear how these two sections (both labeled as “4.2”) are linked with the preceding section 4.1, i.e. how these different types of data (methane isotopes, gas concentrations, fDOM composition, bulk geochemical parameters, biogeochemical environments) are linked. The material covered in these two sections should be either presented in a separate manuscript or the role of the data shown here in the discussion of methane origin (“the hypothesis on subzero methanogenesis”?) should be made more explicit.
Overall, this version of the manuscript reads more like a data report rather than a scientific contribution that clearly identifies what knowledge gaps it tries to address and how the results of this work would contribute to reducing those gaps.
OTHER ISSUES
Lines 164-178
What kind of certified standards were used to calibrate methane d13C and dD data? How was the performance of the Delta S and MAT 253 instruments monitored during the isotopic measurements? How the precession of sample measurements was assessed, i.e. based on how many replicates is the uncertainty reported? What was it for the d13C measurements?
Lines 194 and 205
There are two sections labelled “2.5”.
Lines 219-220, “Methane concentrations were ranging … (Table 1)”
Table 1 does not show any data on methane concentrations. Is this supposed to be Table S1? If so, the link shown on line 505 is not working. Please clarify.
Line 247, “The values of isotope separation (alpha)” here and throughout the manuscript
The term “fractionation” should be used instead of “separation”.
Line 279, the title of section 3.4 “Molecular composition of DOM” and the title of Table 4
The data presented here deals mainly with different fractions of organic matter rather than molecular composition. As such, a more appropriate name for the section (and the table) would be “DOM fractions”
Lines 385 and 432
There are two sections labelled “4.2”.
Lines 373-374, “According to the geochemical data”
What are these data? Please specify.
Lines 479, “Close values of hydrogen isotope partition between CH4 and formation H2O”
It is not clear what the authors mean by “partition”. Is this supposed to be “fractionation”?
Lines 516-518, the ‘Acknowledgements’ section
It looks like the authors have left the template text without adding any information to it.
FIGURES
Figure 3.
What do the numbers in circles refer to? Specify this in the caption for the figure.